# Generating dynamic gene expression patterns without the need for regulatory circuits

**Sahil B. Shah**[iD]**, Alexis M. Hill, Claus O. Wilke**[iD]**\*, Adam J. Hockenberry**[iD]**\***

Department of Integrative Biology, The University of Texas at Austin, Austin, TX, United States of America

\* wilke@austin.utexas.edu (COW); adam.hockenberry@utexas.edu (AJH)

**Data Availability Statement:** All code and data required to reproduce this work are available at: https://github.com/SahilBShah/pinetree-evolution and https://doi.org/10.5281/zenodo.4592577.

## Abstract

Synthetic biology has successfully advanced our ability to design and implement complex, time-varying genetic circuits to control the expression of recombinant proteins. However, these circuits typically require the production of regulatory genes whose only purpose is to coordinate expression of other genes. When designing very small genetic constructs, such as viral genomes, we may want to avoid introducing such auxiliary gene products while nevertheless encoding complex expression dynamics. To this end, here we demonstrate that varying only the placement and strengths of promoters, terminators, and RNase cleavage sites in a computational model of a bacteriophage genome is sufficient to achieve solutions to a variety of basic gene expression patterns. We discover these genetic solutions by computationally evolving genomes to reproduce desired gene expression time-course data. Our approach shows that non-trivial patterns can be evolved, including patterns where the relative ordering of genes by abundance changes over time. We find that some patterns are easier to evolve than others, and comparable expression patterns can be achieved via different genetic architectures. Our work opens up a novel avenue to genome engineering via fine-tuning the balance of gene expression and gene degradation rates.

## Introduction

All genomes encode a variety of distinct RNA and protein sequences, and the production of these biomolecules is essential for the growth and survival of organisms. Critically, these various gene products are often required in different amounts relative to one-another and this demand further varies over time [1–3]. Thus, organisms must have the ability to modulate gene expression levels to meet lifestyle needs and adapt to changing environmental conditions [4–6]. Many gene-regulatory elements are well characterized and cells have been directly engineered to produce individual protein products for decades; the strength of promoters, terminators, ribosome binding sites, *etc.* can all vary over several orders of magnitude to produce different amounts of recombinant gene products [7–9].

More recently, the field of synthetic biology has developed and achieved success in engineering large, time-varying genetic circuits that are characterized by complex interactions and

**Funding:** S.B.S received support from the Texas Institute for Discovery Education in Science (TIDES) in the College of Natural Sciences at the University of Texas at Austin. C.O.W. was supported by a National Institutes of Health grant R01 GM088344, as well as support from the Jane and Roland Blumberg Centennial Professorship in Molecular Evolution and the Dwight W. and Blanche Faye Reeder Centennial Fellowship in Systematic and Evolutionary Biology at The University of Texas at Austin. A.J.H. was supported by National Institutes of Health award F32 GM130113. The Texas Advanced Computing Center (TACC) at The University of Texas at Austin provided high-performance computing resources. The funders had no role in study design, data collection and analysis, decision to publish, or preparation of the manuscript.

**Competing interests:** The authors have declared that no competing interests exist.

often require producing gene products whose sole job is to regulate the expression of other genes [10–15]. While the scale and capabilities of these applications are continually expanding [16–18], the goal of designing entire genomes from scratch presents tremendous challenges. One particular source of challenges is genome complexity—even the smallest free living organisms encode hundreds of gene products that interact with one-another and alter gene expression patterns in ways that are difficult to predict [19].

Even smaller model systems are viruses and bacteriophages. Bacteriophages are useful from an engineering and genome-design standpoint because of their comparatively small size, genetic tractability, and potential uses in medical and biotechnological applications [20–25]. Despite the limited set of genes encoded by their genomes, phages are nevertheless capable of producing complex, time-varying patterns of gene expression [26]. While some of these dynamics are governed by networks of sequence-specific promoters and transcription factors, some phages—such as the *Escherichia coli* phage T7—appear to regulate a portion of their transcriptional demand by tuning production and degradation rates [27, 28]. The full range of expression dynamics that can be achieved by varying only these mechanisms is currently unknown.

Here, we use an *in silico* model of phage infection to explore the range of gene expression complexity that can be achieved using only basic regulatory mechanisms, including transcription, transcript termination, transcript cleavage, and transcript degradation. Our approach relies on computationally evolving genomes with variable-strength promoters, terminators, and RNase cleavage sites to reproduce predefined gene expression time-course patterns. We show that this molecular-level simulation framework can discover solutions to numerous nontrivial patterns and that successfully evolved genomes for particular patterns display a variety of distinct genome architectures. Taken together, these findings lay the groundwork for future efforts to use *in silico* evolutionary design methods to engineer novel phage genomes.

## Materials and methods

### Gene expression simulations

We used the gene expression simulation platform Pinetree [29], version 0.3.0, to simulate expression of mRNA transcripts encoded by a bacteriophage genome containing a small number of genes (between three and ten in our simulations). Our gene simulations included mRNA translation and protein production—with equal ribosome binding site strengths assigned to all genes—but for the purposes of this study we analyzed only mRNA transcript levels. Gene expression in Pinetree is stochastic, and it is modeled at the level of individual molecules. In particular, RNA polymerases are tracked individually as they attach to the bacteriophage genome and commence transcription one nucleoside at a time. Additionally, the polymerases can: i) collide with other polymerases transcribing the same genomic region, ii) terminate transcription upon encountering a terminator element, or iii) read through a terminator.

In addition, RNase molecules can attach to RNAse cleavage sites within transcripts and cleave them—leading to the subsequent directional degradation of RNA fragments from the 5' to the 3' end. Degradation of transcripts from cleavage sites is 1000-fold more efficient than degradation of nascent transcripts from their 5' end. A detailed explanation and justification for the directional transcript degradation model is provided in Ref. [28]. We additionally extended this prior work to allow for site-specific RNase cleavage strengths, and included this feature in the Pinetree 0.3.0 release. RNase cleavage strengths can be thought of as composite parameters that incorporate both the binding specificity of the RNase molecule and the

enyzmatic cleavage rate. Either of these biochemical processes may exhibit sequence specificity, and here we did not model them individually.

Pinetree takes an arbitrary genome as input (which we note is not a sequence file but rather an input file defining the location of individual genes, promoters, terminators, etc.), as well as parameters describing rate constants and cellular-level properties of the infected host cell, such as cell volume or number of specific molecules within the cell. We kept most values at their defaults, but changed the footprint sizes of polymerase and ribosome molecules to better reflect their realistic size (35 and 30, respectively) and additionally set the polymerase copy number to a value of 4 to shorten simulation run-times. These polymerase molecules are all previously existing host-cell polymerases that do not degrade and are being co-opted by the phage. However, we note that the Pinetree platform allows for the production of phage-specific polymerases (provided that they are encoded on the phage genome) and this flexibility may allow for the production of more complicated genetic circuits—which are beyond the scope of this work. In all cases, we simulated the initial five minutes of real time infection.

The primary phage model we studied here consists of genomes containing three protein-coding genes, each 150 nucleotides long. Thus, each encoded protein was 50 amino acids long, and none of the proteins had any function within the simulation; we merely used them to study arbitrary gene expression regulation. We also considered genomes containing ten protein-coding genes, to investigate the scalability of our approach to larger systems. In all cases, the complete genome is available for all molecular binding events at the beginning of the simulation.

## Evolving genomes with specific gene expression patterns

We used our stochastic model of gene expression to evolve phage genomes to display specific temporal patterns of expression. We chose to implement a stochastic, molecular-level simulation because ordinary differential equation-based models make a number of simplifying assumptions and thus are unable to accurately model important biological effects such as molecular collisions. In all cases, we first defined a target phenotype as a gene expression time-course for all genes included in the genome. We then took a starting genome with a single, moderate-strength promoter upstream of the first coding sequence in the genome, and we subjected this genome to subsequent rounds of mutation and selection. Our evolutionary simulations ran for 5000 generations each, and we implemented a simulated annealing procedure that steadily increased the strength of selection with increasing number of generations. The goal with this setup was not to simulate realistic evolutionary processes of natural populations but rather to permit efficient exploration of a potentially rugged fitness landscape and thus maximize our chance of finding suitable genome architectures.

We considered three different types of mutations: addition of a regulatory element (in this case, either a promoter, terminator, or RNase cleavage site), removal of a single existing regulatory element, or modification of the binding constant for an existing regulatory element. The addition and removal functions act as insertion and deletion mutations, respectively, whereas modifications to binding strengths are analogous to point mutations that alter specific rate parameters. All protein-coding sequences are defined in the starting genome and are not altered through any mutational step.

To determine the appropriate rate constants for individual regulatory elements as well as their starting values when they are inserted into a genome, we simulated simplified genomes and measured the expression response of downstream genes (S1 Fig in S1 File) using a model containing only three coding sequences. From this qualitative analysis we chose to insert promoters at a strength of $10^6$ (with minimum and maximum values set at $10^5$ and $10^{13}$,

respectively), RNAse sites at $5 \times 10^{-3}$ (with minimum and maximum values set at 0 and 1), and terminators at a strength of 0.2 (with minimum and maximum values set at 0 and 1). After inserting an element, the element strength can be modified in subsequent mutational steps. Each modification is proposed by drawing a value from a normal distribution with mean of 1 and standard deviation of 0.1, and multiplying this value by the current element strength. If the proposed mutation results in an out-of-bounds value, the value is simply discarded and the process repeated until a valid proposed mutation is drawn.

Our mutational process is not meant to mimic natural evolution. Rather, *some* mutation is always proposed at every generation—either element addition, removal, or strength modification. This is accomplished by first enumerating all possible mutations given the current genome status (deletion and modification can happen to every existing element, and every non-existing element can be added) and then randomly selecting one of the possibilities from this set (*i.e.* addition of a promoter between genes 2 and 3). Only one of each element-type can exist in a particular region between two genes, so once there is a terminator between genes 2 and 3, for instance, another terminator can not be proposed in this specific location.

While some mutation will always be proposed at each generation, many of these mutations will not be accepted. We calculated the fitness of a genome from the distance between its gene expression pattern and the target pattern. Because gene expression patterns are obtained from stochastic simulations, we performed ten replicate simulations for each genome and averaged mRNA abundances across replicates before proceeding with the fitness calculations and determination of acceptance probabilities. We defined fitness in terms of the normalized root mean square error (normalized RMSE). We obtained normalized RMSE by first calculating the RMSE for a single gene $k$,

$$\mathrm{RMSE}_k = \sqrt{\frac{\sum_{t=1}^{T} \left[y_k(t) - Y_k(t)\right]^2}{T}}, \tag{1}$$

where $y_k(t)$ is the observed gene expression level of gene $k$ at time $t$, $Y_k(t)$ is the target gene expression level of gene $k$ at time $t$, and $T$ is the total number of time steps simulated for the gene expression time course. Subsequently, we normalized each $\mathrm{RMSE}_k$ by the mean target abundance, $\bar{Y}_k = \sum_{t=1}^{T} Y_k(t)/T$, and then averaged these values across all genes,

$$\mathrm{RMSE}_{\mathrm{norm}} = \sum_{k=1}^{M} \mathrm{RMSE}_k/(M\bar{Y}_k), \tag{2}$$

where $M$ is the total number of genes in the genome. Additionally, and after analysis of several distinct patterns and simulations, we determined that normalized-RMSE values $\leq 0.1$ aligned with our expectation about the suitability of individual solutions. We settled on this value and applied this threshold throughout the results whenever we refer to simulations that produce a final pattern that successfully matched the target.

To convert $\mathrm{RMSE}_{\mathrm{norm}}$ into fitness $f$, we employed the Fermi function

$$f = 1/[\exp(\beta \ \mathrm{RMSE}_{\mathrm{norm}}) + 1]. \tag{3}$$

We have $f = 1/2$ when $\mathrm{RMSE}_{\mathrm{norm}} = 0$ (*i.e.* when the genome exactly displays the target expression pattern) and $f$ monotonously declines to zero as $\mathrm{RMSE}_{\mathrm{norm}}$ increases. The constant $\beta$ determines how quickly $f$ declines with increasing $\mathrm{RMSE}_{\mathrm{norm}}$, and thus we can vary $\beta$ to modify the strength of selection in the simulations.

In addition to the Fermi function, we also explored two additional fitness functions, exponential decline,

$$f = \exp(-\beta \; \text{RMSE}_{\text{norm}}), \tag{4}$$

and linear decline,

$$f = -\beta \; \text{RMSE}_{\text{norm}}. \tag{5}$$

Both of these functions produce qualitatively similar results to the Fermi function.

We evolved genomes using an accelerated origin–fixation model [30]. In this model, at any point in time the evolving population is represented by a single resident genotype. Whenever a novel, mutated genotypes arises, it can either go to fixation, thus becoming the next resident genotype, or it can be lost to genetic drift. Whether a new genotype is accepted or not is determined probabilistically according to the value of $P_{\text{accept}}$, defined as

$$P_{\text{accept}} = \begin{cases} 1 & \text{if } f' \geq f, \\ (f'/f)^{2N_e} & \text{if } f' < f, \end{cases} \tag{6}$$

where $f$ is the fitness of the resident genotype, $f'$ is the fitness of the new mutant, and $N_e$ is the simulated effective population size. We note that beneficial mutations are unconditionally accepted. This choice allows for rapid exploration of the fitness landscape while resulting in steady-state sampling identical to the one obtained when using the traditional Kimura formula for the probability of fixation [30].

To allow for even more rapid sampling of the fitness landscape, we implemented a simulated annealing procedure where selection was weak initially but became increasingly stronger towards the end of the evolution. We modified the strength of selection by varying the parameter $\beta$ in Eq (3), while keeping $N_e$ constant at $N_e = 1000$ throughout. We varied $\beta$ according to the following schedule: For the first 500 generations, $\beta = 10^{-3}$, to noisily explore the fitness landscape (S2 Fig in S1 File). From generation 501 to 4500, we linearly increased $\beta$ from $10^{-3}$ to 1.1. Finally, from generation 4501 to 5000, we linearly increased $\beta$ from 1.1 to 1.3. This simulated annealing approach ensured that evolutionary simulations explored a wide range of distinct genome architectures yet tended to settle on one genome architecture towards the end of each evolution. For each pattern that we explored, we performed 50 replicate simulations using the above approach.

## Accounting for possible gene re-arrangements

In our gene expression model, gene order may affect observed gene expression levels because transcription proceeds from the 5' end to the 3' end but may stochastically terminate at defined terminator sequences (with varying efficiency). Therefore, gene identity matters when comparing a genome's expression pattern to a target pattern and it is a very different requirement to state that the first gene in a genome has to be the most expressed versus the last gene.

In practical applications of genome engineering, however, we would not normally be concerned about the order of genes in the genome, as long as each gene is expressed at appropriate levels over time. Thus, we explored *general patterns* by simulating all possible gene arrangements for each pattern. For a three-gene genome, there are six possible combinations for matching the genes in the genome to the genes described in the target expression pattern. We simulated all six variations for each pattern, and then we selected the gene arrangement that had the lowest normalized-RMSE (averaged across 50 independent simulation replicates) as the representative for that pattern. For the 10 gene simulation, we simulated only a single predetermined gene order and note that attempting to simulate all possible gene order variants would be impractical for this case.

## Removing elements with negligible effect

When analyzing final evolved genome architectures, we wanted to be certain that each regulatory element (promoter, terminator, RNase cleavage site) contained within that genome had a substantial effect on the overall expression pattern. We did this solely so that we could investigate individual genome architecture solutions and compare the diversity of genome architectures that evolved for particular patterns. Since rate constants for elements were allowed to evolve, it was possible that individual elements evolved towards rate constants that were so low that the elements had essentially no function. Thus, at the end of evolutionary simulations, we evaluated each element's effect on the final gene expression pattern and removed all elements with a negligible effect.

To do so, we removed each element from the genome and then analyzed how the resulting gene expression pattern changed relative to the complete genome as it evolved. If the normalized-RMSE value remained below the threshold that we used as a quantitative indicator of a successful simulation (using a threshold of 0.1, as previously noted), we recorded the element as a candidate for removal. After evaluating all regulatory elements in this manner, we removed the element whose removal had the least impact. We then repeated this process, greedily removing elements one by one until no further elements could be removed without pushing the normalized-RMSE value above our threshold of success. We note that this greedy algorithm for removing individual elements neglects potential epistatic effects, such as when different regulatory elements have compensatory or conflicting roles. Given the small size of the genomes that we focused on here, we expect that these effects are generally minor but may nevertheless become more pronounced for larger and more complex genomes.

## Assessing the diversity of evolved genomes

Our evolutionary simulations generally resulted in multiple, independently evolved genomes that had different genome architectures for each target pattern. To quantify the diversity of evolved architectures, we calculated the entropy of the set of evolved solutions that were deemed successful for the given target. In this calculation, we only considered the presence or absence of individual regulatory elements (after removing insubstantial elements, as described above) in the genome while ignoring the rate constants/element strengths. We identified the unique genome architectures among the $n$ successful solutions (ignoring simulations that produced a normalized-RMSE $>0.1$) and then determined the count $n_i$ for each unique architecture $i$ (so that $\sum_i n_i = n$). We then calculated entropy $H$ as

$$H = -\sum_i (n_i/n)\log_2(n_i/n). \tag{7}$$

Because the logarithm is taken to base 2, the resulting entropy value is measured in units of bits.

## Data and code availability

All code and data required to reproduce this work are available at: https://github.com/SahilBShah/pinetree-evolution and also archived on Zenodo at https://doi.org/10.5281/zenodo.4592577.

# Results

## Evolutionary simulation of phage gene expression

Our goal is to understand the range of possible gene expression patterns that a phage could produce by varying only a small set of regulatory components. We reasoned that rational

genome design would be difficult for all but the most trivial cases, and that fully enumerating all possible genomes is impractical. Instead, we focused on developing an evolutionary strategy to engineer *in silico* genomes to match a diverse set of target pre-defined gene expression time-courses.

This computational strategy that we developed relies on a stochastic gene expression platform (Pinetree), which simulates the process of phage infection in molecular-level detail and produces time-course data of RNA abundance as an output [29]. We emphasize here that Pinetree does not operate directly on DNA sequences but rather uses parameter file inputs that define the location and strength of individual genes, promoters, terminators, etc. on a hypothetical genome sequence. We built upon this software to enable evolutionary simulations, where discrete generations consist of individual simulations of phage infection and the resulting RNA species time-course is our ultimate phenotype of interest. Pinetree expects a genome as input and under our evolutionary approach this genome varies from generation-to-generation as the location and strengths of promoters, terminators, and RNase cleavage sites are subject to mutation, selection, and drift.

As a proof-of-principle, we first used Pinetree to simulate gene expression from an arbitrary example genome architecture. The positive control genome architecture and resulting gene expression pattern produced from a single simulation are shown in Fig 1A (first panel). Using this gene expression output as our target data, we next attempted to evolve a genome from scratch that was capable of producing similar levels of gene expression over time. Our evolutionary simulation starts with a three-gene genome containing only a single promoter, whose

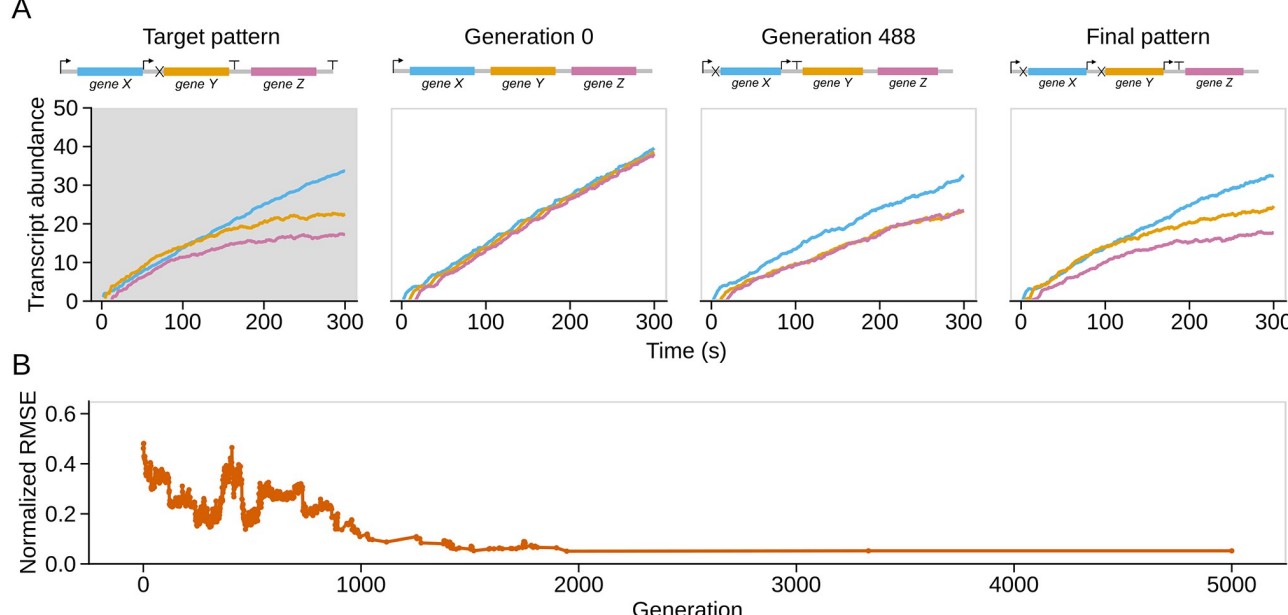

**Fig 1. Evolutionary simulation of a positive control gene expression pattern.** A: Genome architectures and corresponding gene expression time courses. The target pattern (first panel) is a gene expression time course generated by an arbitrarily chosen genome architecture simulated in Pinetree. This architecture has a promoter in front of genes X and Y (indicated by inverted L with an arrow pointing right), a transcriptional terminator after genes Y and Z (indicated by a T), and an RNase cleavage site in front of gene Y (indicated by a cross). Line colors correspond to the genes labeled in the genome architecture. We use these symbols and colors consistently throughout this work to display genome architectures. The remaining panels illustrate the evolutionary process: the starting genome architecture and corresponding gene expression time-course (generation 0), an example from the middle of the evolutionary simulation (generation 488), and the final evolved architecture (final pattern). The final evolved genome architecture is similar but not identical to the target. B: The normalized-RMSE metric declines with increasing numbers of generations as the genome architecture evolves to reproduce expression patterns that are increasingly similar to the target pattern.

time-course of gene expression initially bears little resemblance to the target. As generations progress, single mutations are proposed and conditionally accepted based on whether they alter the resulting gene expression pattern to more closely resemble the target (see Materials and methods). More specifically, we define the fitness of a mutation as the inverse of the normalized-Root Mean Squared Error (RMSE) relative to the target. Individual mutations that result in a better match to the target expression pattern (*i.e.* have lower normalized-RMSE values) are accepted, with some probability of accepting slightly deleterious mutations to allow for efficient exploration of the potentially rugged fitness landscape.

As shown in Fig 1A (right-most panel), the best gene expression time-course that evolved over a 5,000 generation evolutionary simulation qualitatively matches the target. To quantitatively make this determination, we found—through visualization of numerous simulations—that normalized-RMSE values ≤ 0.1 generally aligned with our qualitative determination about the success of any given solution. We henceforth consider an evolutionary simulation successful if the final normalized-RMSE falls below this value, which occurs in this case (Fig 1B). Thus, this positive control shows that our approach is capable of evolving genome architectures that match complex patterns from simple starting points.

## Evolving genomes to match a range of gene expression patterns

After successfully conducting an evolutionary simulation with a positive control, we applied the same method to a range of gene expression patterns that have no *a priori* known genetic solution. We began with a simple pattern where all three genes linearly increase in abundance over time but do so at different rates (Fig 2A, left two panels). A rational solution to this pattern might be to have a strong upstream promoter with weak terminators between each successive gene. An alternative solution would be to start with a weak upstream promoter followed by subsequent promoters between each gene so that each successive gene will be expressed at a higher level. While in the first case that we outlined the *first* gene in the genome would be

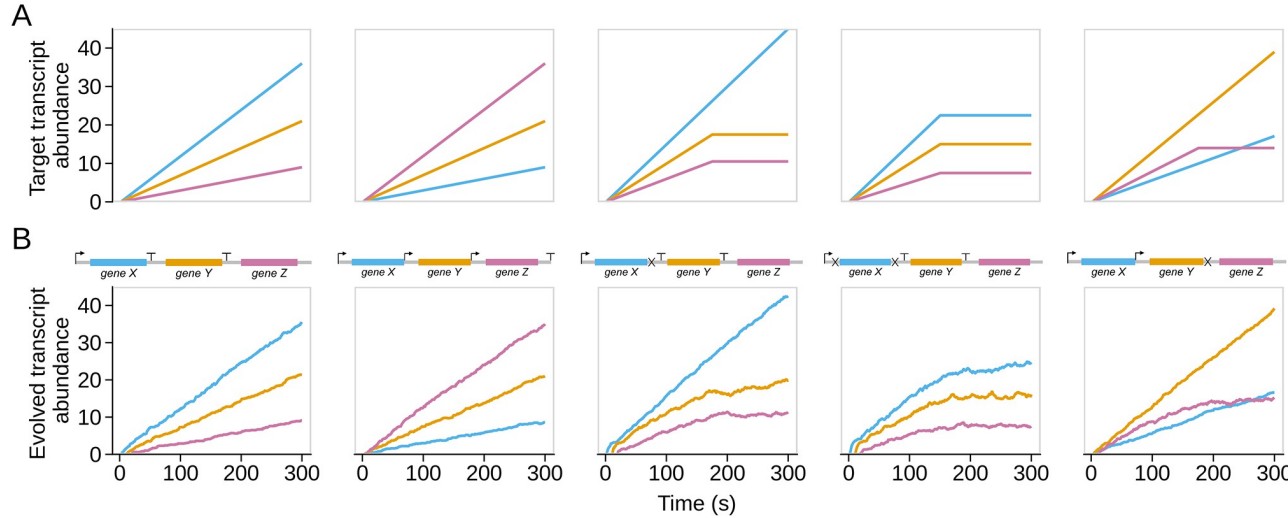

**Fig 2. Examples of successfully evolved gene expression patterns.** A: Each panel contains a target gene expression time-course pattern. Collectively, these patterns span a diverse range of possibilities. B: The corresponding representative gene expression patterns from successful simulations for each respective target. Shown above each time-course of gene expression is the evolved genome architecture that produced it. The architectures in the first two panels contain only promoters and terminators, while the remaining three architectures also contain RNase cleavage sites. RNase cleavage sites seem to be required for any regulatory patterns that go beyond a simple linear increase in expression level over time.

expressed at the highest rate, in this latter scenario the *last* gene would be expressed at the highest rate.

As we expected, the left two panels of Fig 2B show that we were able to successfully evolve this linearly increasing pattern (with two different gene arrangements) and that the evolved genome architectures match our rational expectation. The final three examples presented in Fig 2 further show successful simulations (the lowest normalized-RMSE values found across 50 independent replicates) for more complex patterns that are difficult to rationalize solutions to. While this demonstrates that genomes in our system can evolve to match several distinct patterns with qualitatively successful solutions, we wanted to further increase the complexity of our targets and evaluate replicate simulations more quantitatively.

We conducted a total of 300 independent evolutionary simulations for each of 10 general patterns (Fig 3A), split into 50 simulations for each of the 6 possible gene arrangements per pattern (S3 Fig in S1 File, see Materials and methods). The patterns we explored were characterized by increases in absolute transcript abundances at various rates, as well as plateaus in abundance that are indicative of a steady-state balance between transcript abundance and degradation. More complex gene expression dynamics—such as gene expression delays and oscillations—will almost certainly require more complex regulatory circuitry than we allow for here. However, the range of possible dynamics that can be achieved even with combinations of plateaus and linear increases is unknown, which is why we chose to focus our efforts on these initial patterns.

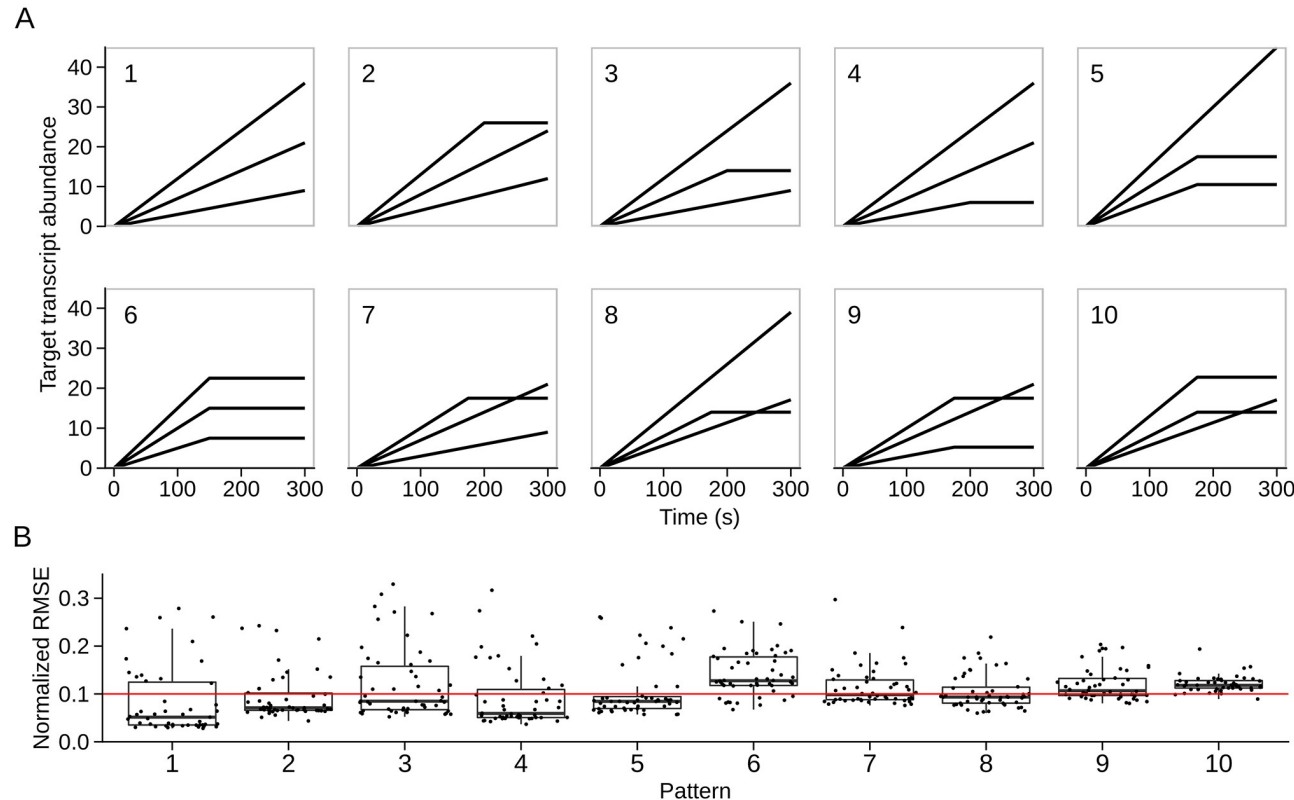

**Fig 3. Quantitative assessment of pattern achievability.** A: The 10 target gene expression time-course patterns that we simulated are displayed as black lines representing general patterns (each of which has 6 possible gene arrangements). B: For 50 replicate simulations of each pattern, shown are the lowest achieved normalized-RMSE values. All patterns had at least two successful independent evolutionary simulations, as determined by the red line highlighting a normalized-RMSE value of 0.1.

For the particular gene arrangement that yielded the best results for each pattern, Fig 3B shows the distribution of normalized-RMSE values from the 50 replicate simulations (S4 Fig in S1 File shows corresponding gene expression time-courses for the best replicate). Each pattern had at least two successful replicate simulations (normalized-RMSE $\leq$ 0.1), but there was nevertheless a clear difference between how easily solutions to particular patterns were able to evolve—compare and contrast boxplots for patterns #5 and #6 in Fig 3B. For pattern #5, the box lies entirely below the red line, indicating that over 75% of simulations were determined to be successful according to our cutoff criterion. By contrast for pattern #6, the box lies entirely above the red line, indicating that over 75% of the simulations were not determined to be successful.

These result suggests certain gene expression time-course patterns may have a more restrictive set of genome architectures capable of producing them, but we also note that the evolutionary parameters could perhaps be altered to better optimize solutions to particular patterns. As a robustness check, we performed evolutionary simulations to match the same 8 patterns depicted in Fig 3 using two additional fitness functions and observed qualitatively similar results (S5 Fig in S1 File).

## Disparate genome architectures can recapitulate the same target expression pattern

Beginning with our positive control simulations (Fig 1), we saw evidence that different genome architectures could produce comparable time-course patterns of gene expression. In this case, while the evolved genome produced a nearly identical gene expression pattern to the target, the evolved genome differed from the input genome that was used to produce the target expression data. To investigate this trend more broadly and for different gene expression patterns, we interrogated the diversity of the successfully evolved genome architectures for each pattern. Overall, we found that there was no single genome architecture that completely dominated results for any of the target patterns. As illustrated in Fig 4A, even the simplest pattern that we assessed had numerous distinct (albeit qualitatively similar) genome architectures that evolved, each of which were capable of producing gene expression time-courses that successfully matched the target. However, some patterns were more heterogeneous than others in terms of the number of possible genome architectures found (Fig 4B).

We quantified the diversity of evolved genome architectures for successful replicate simulations for each pattern using information entropy (as in Fig 3 we analyzed data from only the best of the 6 possible gene arrangements for each pattern). Lower entropy values imply that only a single or a small number of distinct genome architectures evolved for a particular pattern. By contrast, high entropy values imply that the genome architecture solutions for a particular pattern were largely distinct from one another. Fig 4C illustrates the entropy scores for each of the 10 patterns showing that pattern 5 had the most diverse set of genome solutions whereas pattern 10 (which had only two successful simulations) was the most constrained. Taken together, these results show that particular patterns may be more flexible than others in terms of the number of possible solutions but the cause of this variability is unknown at this stage and a possible avenue for future research.

## Evolutionary simulation of a ten-gene model

Although the three-gene model is ideal for studying the basic principles behind genetic regulation, the size of this genome is unrealistically small compared to most phage genomes. Thus, we wanted to test whether our simulation approach was capable of evolving larger genomes containing up to 10 genes (on par with the smallest known phage genomes). As shown in Fig

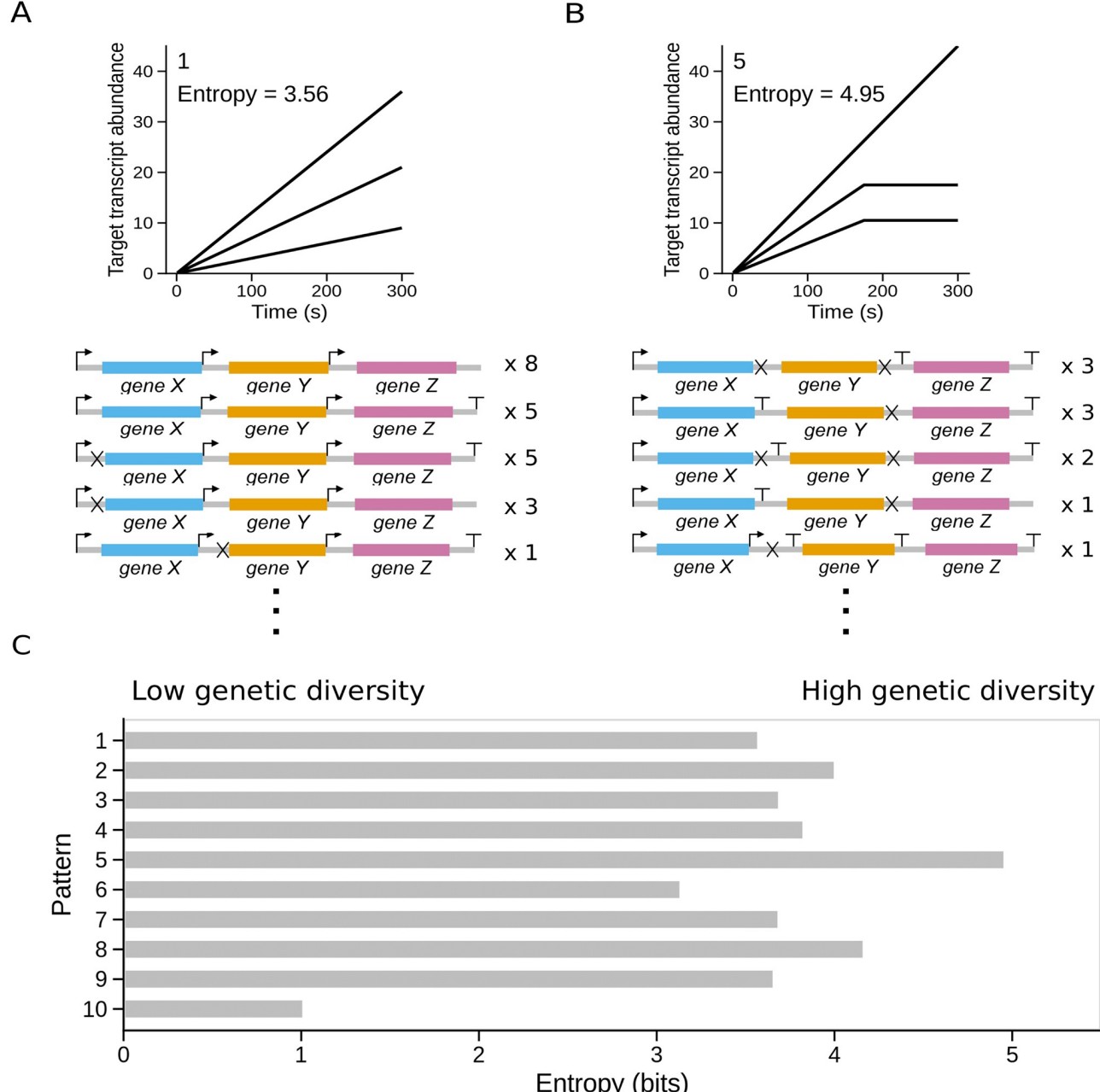

**Fig 4. Diversity of successfully evolved genome architectures.** A: A sample of the successfully evolved genome architectures for pattern #1 showing that multiple potential architectures can reproduce the target expression pattern. Only the architectures that evolved most frequently are shown. Multiples on the right indicate how often each architecture arose among 50 replicates. B: Similar to panel (A), a sample of the successfully evolved architectures for pattern #5. We note the much higher frequency of terminators and RNase cleavage sites in genome architectures evolved for pattern #5 versus pattern #1. C: Entropy values for the set of successfully evolved genome architectures for each target pattern. Lower values indicate that successful replicate simulations converged upon one or a small number of genome architectures. More specifically, an entropy value of 1 corresponds to their being effectively 2 ($2^1$) distinct architectures for a given pattern from amongst the replicates, while an entropy value of 5 corresponds to their being effectively 32 ($2^5$) distinct architectures.

5, we achieved a qualitatively suitable result when attempting to simulate a simple linear pattern with variable rates of increase for each gene within the genome. However, we note that increasing the number of genes within the genome also increases the run-time of the simulation substantially since more generations are required to adequately explore the fitness landscape given that there are many more possible mutations (*i.e.* regions between genes that can potentially contain regulatory elements). This result, nevertheless, shows that our approach is generalizable and can, in principle, be extended to any number of genes with complex, predefined target patterns barring computational limitations.

A further feature that is evident within this simulation is that the 3' most genes experience significant expression delays, presumably as a result of having evolved weaker promoters that are out-competed for polymerase molecules by the stronger 5' promoters. However, even within this 10-gene system the time-scale of these delays is modest in comparison with the 5 minute simulation time. Implementing substantial gene expression delays into realistic systems that rely solely on this delay-feature would require inserting large pieces of non-coding DNA to space apart the temporal expression of genes. While this is certainly possible, we find it likely that substantial gene expression delays can be more efficiently implemented via genetic interactions that we do not consider here.

## Summary of observed design patterns

From the entirety of our simulation results, we have identified the following patterns of regulatory control. In general, combinations of promoters and terminators only lead to a linear increase of transcript abundances, possibly at differing rates, as the effect of these regulatory elements does not depend on the current abundance of a transcript. For instance, for the first gene to be expressed more rapidly than the second gene, there can be a strong promoter in front of the first gene and a weak terminator in front of the second gene. Examples include the first column of Fig 2 and also Fig 5. By contrast, a weak promoter in front of the first gene and a second weak promoter in front of the second gene will create a pattern where the second gene is expressed more rapidly than the first gene (second column of Fig 2). RNase cleavage sites are required for time courses where transcript levels reach a steady state, as the amount of degradation following cleavage is proportional to transcript levels and thus increases with increasing transcript abundance (examples include the three right-most columns in Fig 2 and all genome architectures in Fig 4B).

For time courses with a cross-over event, where one transcript levels off while another, initially less abundant transcript keeps steadily increasing (last column in Fig 2), we have found that typically a weak promoter precedes the first gene, a strong promoter precedes the second gene, and an RNase cleavage site also precedes the second gene but follows the second promoter. The weak promoter allows for the first gene to steadily increase at a low rate. The strong promoter rescues transcription of the downstream gene. The following RNase cleavage site will degrade the transcripts produced, in proportion to the transcripts' abundance, and thus the transcripts of the second gene will eventually reach steady state.

Finally, even though RNase cleavage sites will generally cause eventual steady state, we have also observed cases where a weak RNase cleavage site follows a strong promoter, resulting in a time course that does not meaningfully level off on the time scale of our simulation results (examples include the third, fourth, and fifth genome architecture in Fig 4A). In those cases, the RNase cleavage site simply serves to reduce the effect of the promoter. This observation highlights that the transcript level at which a steady state is reached depends on the relative strengths of the promoter and the RNase cleavage site, and for sufficiently strong promoters and weak cleavage sites a steady state may never be observed during physiological simulation

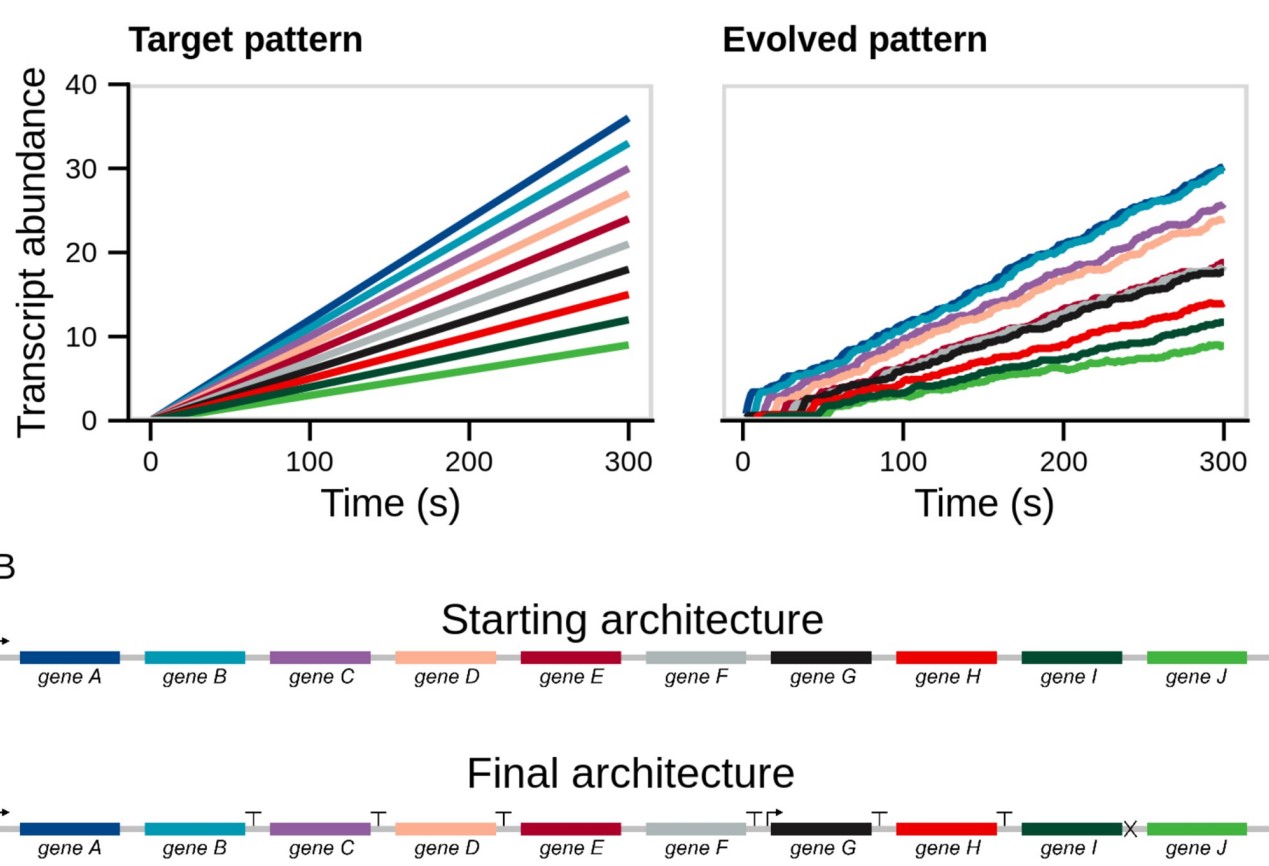

**Fig 5. Evolutionary simulation of a ten-gene model.** A: Shown are the target (left) and evolved (right) patterns. The evolved pattern is the best of 5 independent replicates, each of which was simulated for 100,000 generations. B: The starting (top) and best fitting evolved (bottom) genome architectures. Differential gene expression in the evolved genome happens primarily via transcription termination with readthrough. The genome has only two promoters and one RNase cleavage site, but six terminators that successively reduce transcription as we move from the 5' to the 3' end of the genome.

times, even though it exists and would be reached if we let the simulations run for sufficiently long times.

## Discussion

Genomes of all species employ a myriad of regulatory strategies to ensure that individual genes are expressed at particular levels that vary over time. Here, we have used computational modeling to demonstrate that bacteriophages can generate a range of complex and time-varying expression patterns without the need for specific regulatory molecules or complex genetic circuitry. By varying only the strength of promoters, terminators, and RNase binding sites, our *in silico* evolution platform was able to generate genomes capable of matching numerous and distinct gene expression time-course patterns. These findings show that networks of interacting promoters and transcription factors are not explicitly necessary to achieve certain gene expression designs, and provide a proof-of-principle for the *de novo* design and engineering of phage genomes using molecular-level evolutionary simulation.

Our study is motivated by the observation that naturally evolved phages appear to regulate their gene expression via the mechanisms discussed here. Phage genomes are typically small

and highly gene dense, which is likely to limit the overall complexity of their native gene expression programs [31–34]. Even with relatively small genomes, numerous studies have shown that phages are capable of producing complex gene expression dynamics over the course of an infection cycle. In phage T7, for instance, genes that are expressed early in the infection cycle are typically involved in shutting down synthesis of host-cell macromolecules, whereas the genes that are expressed later have roles in viral assembly and lysis [35–37]. While the T7 genome encodes its own polymerase that is critical for the delayed expression of these late genes, phage T7 gene expression dynamics are partially accomplished through the use of variable strength promoters, transcriptional terminators, and RNase cleavage sites [27, 28, 38–41]. Incorporating knowledge of variable strength regulatory elements has been critical for generating accurate computational models of phage T7 infection that are increasingly able to recapitulate empirically measured gene expression abundances [29, 38, 42, 43]. The findings presented here suggest that this level of regulatory control is likely to apply more broadly to phages in general.

The arrangements of regulatory elements that arose in our simulations look similar to what is observed in bacteriophage genomes. For example, the T7 genome contains 10 RNase III cleavage sites, six of which co-occur in the same intergenic region as a promoter [44]. This arrangement of promoters and cleavage sites results in transcript ramps and cliffs [28], where transcript abundances fall off near cleavage sites, and then steadily increase due to the presence of promoters. Similarly, bacteriophage ΦX174 has a small (5386 nt) circular genome composed of 11 genes, including a cluster of co-localized capsid genes [45]. Capsid gene expression is regulated by four terminators of varying strengths [46], arranged in an alternating gene-terminator pattern similar to that of the evolved ten-gene architecture (Fig 5). In ΦX174, terminator read-through produces a stepwise pattern of decreasing transcript abundances [34], which may help ensure that structural proteins are produced with the stoichiometries necessary for proper capsid assembly. ΦX174 also contains an overlapping promoter-terminator element, situated in the intergenic region between its structural and replication genes. In our evolved genome architectures, there is at least one example of such an element (Fig 1, right two panels); in both our simulation and in ΦX174, this regulatory feature appears to decouple expression of the adjacent genes. Finally, bacteriophage T4, which has a large genome encoding over 300 genes, makes use of at least 119 promoters, as well as a number of terminators and mRNA cleavage sites, many of which co-occur as groups of two or more regulatory elements [47]. Combinations of promoters, terminators, and cleavage sites give rise to the complex, time-dependent gene expression patterns observed in T4, which are further controlled by phage-encoded regulatory proteins.

Our findings here highlight some of the strengths and the limitations of regulatory approaches built solely off varying the strength of transcription, termination, and transcript degradation. The target phenotypes we used were fairly simple. We initially only considered cases in which each gene's transcript abundance increased linearly. We then gradually increased the complexity of each gene's expression pattern by allowing some genes to achieve steady state at certain time points. Such patterns could be evolved but we only observed a few successful simulations among 50 replicates. Our system shows that gene products can either accumulate at different rates over time via variable strength transcription or plateau at different levels by tuning degradation rates. In the most complex cases that we investigated, a combination of variable production and degradation rates was sufficient to produce expression dynamics whereby the relative ordering of genes by abundance could switch over time (Fig 3, patterns 7–10). However, the regulatory elements that we assessed do not provide a means to shut off the production of certain genes, which would enable decreases in *absolute* transcript abundances at particular times. Similarly, gene expression delays within this system are limited

to short time scales; there is no apparent mechanism for inducing expression at a particular time without considering more complex genetic regulation—such as the early production of a phage-specific polymerase that controls later genes as in phage T7. Finally, without allowing for more complex genetic interactions, we find it unlikely that important dynamic features such as oscillations can reliably evolve given the limited set of regulatory elements that we considered here.

Despite these constraints, our approach may provide important insight into the overall organization and regulatory principles governing phage genomes. For instance, RNase cleavage sites and promoter sequences frequently co-occur in phage T7 [28]. In our simulations, we observed numerous instances of this co-occurrence, indicating that this motif may provide a general means for decoupling the expression of proximal genes. However, the generality and significance of this finding is difficult to assess based on the limited set of patterns that we investigated. Future work, particularly focused on larger genomes with more complex expression dynamics, might uncover interesting design principles that have heretofore gone unnoticed.

While our results provide insight into the limits and capabilities of phage regulatory strategies, our approach may also prove useful for synthetic biology applications in phages [48, 49]. Synthetic genetic circuits are typically designed for free-living species where they must operate in variable cellular contexts over comparatively long time-scales [50, 51]. By contrast, phage infections are typically brief, with relevant time-scales on the order of tens of minutes [52, 53]. We performed our simulations with realistic cellular parameters over an initial 5 minute infection period; these parameters were previously tuned to fit empirical data on phage T7 gene expression [28] and thus represent approximately accurate values in terms of molecular rates/abundances, translocation speeds, etc. Producing more complicated dynamics from much larger genomes over longer timescales could, in principle, be accomplished using minimal genetic circuitry to divide the genome into early and late gene sets; within each of these two sets our results show that it is possible to encode diverse dynamics without any further regulatory mechanisms. Indeed, the phage T7 genome is partitioned in precisely this manner, having distinct classes of early, middle, and late expressed genes that perform distinct functions [35–37].

However, in any biological application, we would ultimately want to manipulate protein abundances, and we have here only considered transcriptional regulation. Protein abundances are determined both by transcript abundances and by additional parameters relevant at the translation stage, such as the rate of translation initiation or the speed of translation elongation, which may be modulated by codon usage or mRNA secondary structure [54]. These additional parameters would have to be considered in our simulation model to produce realistic predictions. We note that the Pinetree simulator we used here already has the capability to model translation, by tracking the movement of individual ribosomes along transcripts [29]. It would be straightforward to evolve systems towards target protein abundances rather than transcript abundances, and to allow mutations in translation initiation rates and/or in codon usage bias, so that each gene can evolve unique translation initiation and elongation rates.

In principle, the approach that we presented here could be adapted to explicitly design phage genomes with specified gene expression dynamics but doing so presents several challenges. While Pinetree relies on genome-level information, it does not make use of *sequence*-level information. Rather, elements such as promoters are encoded by their location and strength—not by an explicit sequence. Translating a particular evolutionary design into an actual genome sequence would thus require using standardized parts with known element strengths; these strengths could be calibrated against the values encoded in Pinetree to aid in genome sequence design [55]. In the case of RNase sites, this may be particularly challenging

because while degradation rates are known to vary considerably across genes, the overall rules concerning RNase binding and cleavage are less well understood than the actions of promoters and terminators [56]. However, this outlook is beginning to change [57–59].

We envision that the results presented here can be extended in a number of different directions in future research. First, we only considered non-overlapping gene arrangements in our study but many phage genomes are highly compact and successive genes frequently overlap; this feature might impact gene expression in ways that are difficult to predict but could be assessed using our approach [23, 31]. Second, an obvious way to create more complex gene expression dynamics would be to encode a regulatory protein on the phage genome rather than the arbitrary proteins that we have thus far investigated. Extensions of our framework to consider more genes would likely benefit from such an approach, which could allow for expression delays or decreases in the abundance of particular transcripts over time. Third, we previously noted that gene order can play an important role in achieving particular targets. With only three genes we were able to enumerate all possible gene arrangements, but this approach is intractable for larger genomes and will require an additional mutational step that swaps the placement of genes for larger applications. Finally, we focused our attention on transcript abundances but differences in translation initiation and elongation rates offer further ways to tune gene expression at the level of protein products to produce more complex dynamics that operate over longer-time scales [7, 8, 60–63].

An extraordinary number of phages have been discovered in recent years, but only a small number of these have been interrogated experimentally [64–68]. Computational approaches can help to better characterize the general constraints and principles that affect genome organization and function, and can additionally provide a way to engineer phages based on predetermined design constraints. The design and engineering of whole phage genomes using principles and approaches that we developed here may have a number of future medical and biotechnological applications, including combating antibiotic resistant bacteria [69].

## Supporting information

**S1 File.**
(PDF)

## Acknowledgments

This work made use of high-performance computing resources provided by the Texas Advanced Computing Center (TACC) at The University of Texas at Austin.

## Author Contributions

**Conceptualization:** Claus O. Wilke, Adam J. Hockenberry.

**Formal analysis:** Sahil B. Shah, Adam J. Hockenberry.

**Funding acquisition:** Claus O. Wilke.

**Investigation:** Sahil B. Shah.

**Methodology:** Alexis M. Hill.

**Project administration:** Adam J. Hockenberry.

**Resources:** Claus O. Wilke.

**Software:** Sahil B. Shah, Alexis M. Hill.

**Supervision:** Claus O. Wilke, Adam J. Hockenberry.

**Validation:** Alexis M. Hill.

**Writing – original draft:** Sahil B. Shah, Adam J. Hockenberry.

**Writing – review & editing:** Sahil B. Shah, Claus O. Wilke, Adam J. Hockenberry.

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
