## [Decision Letter · Decision Letter 0]

14 Jan 2022

PONE-D-21-23762Generating dynamic gene expression patterns without the need for regulatory circuitsPLOS ONE

Dear Dr. Hockenberry,

Thank you for submitting your manuscript to PLOS ONE. After careful consideration, we feel that it has merit but does not fully meet PLOS ONE’s publication criteria as it currently stands. Therefore, we invite you to submit a revised version of the manuscript that addresses the points raised during the review process. ==============================Comments from the Academic Editor:Reviewer 1, Reviewer 2, and I all feel that the work presented in this manuscript is both scientifically sound and of interest to the PLOS One reader base. I expect to accept this work for publication in PLOS One after the completion of a minor revision that addresses all of the comments made by Reviewers 1 and 2.

Please submit your revised manuscript by Feb 28 2022 11:59PM. If you will need more time than this to complete your revisions, please reply to this message or contact the journal office at plosone@plos.org. Please include the following items when submitting your revised manuscript:A rebuttal letter that responds to each point raised by the academic editor and reviewer(s). You should upload this letter as a separate file labeled 'Response to Reviewers'.A marked-up copy of your manuscript that highlights changes made to the original version. You should upload this as a separate file labeled 'Revised Manuscript with Track Changes'.An unmarked version of your revised paper without tracked changes. You should upload this as a separate file labeled 'Manuscript'.

We look forward to receiving your revised manuscript.

Kind regards,

William Ott, Ph.D.

Academic Editor

PLOS ONE

Journal Requirements:

“This work was supported by National Institutes of Health grants R01 GM088344 to C.O.W. and F32 GM130113 to A.J.H. C.O.W. also received support from the Jane and Roland Blumberg Centennial Professorship in Molecular Evolution and the Dwight W.and Blanche Faye Reeder Centennial Fellowship in Systematic and Evolutionary Biology at The University of Texas at Austin. S.B.S received support from a TIDES research fellowship. The Texas Advanced Computing Center (TACC) at The University of Texas at Austin provided high-performance computing resources.”

“S.B.S received support from the Texas Institute for Discovery Education in Science (TIDES) in the College of Natural Sciences at the University of Texas at Austin. C.O.W. was supported by a National Institutes of Health grant R01 GM088344, as well as support from the Jane and Roland Blumberg Centennial Professorship in Molecular Evolution and the Dwight W. and Blanche Faye Reeder Centennial Fellowship in Systematic and Evolutionary Biology at The University of Texas at Austin. A.J.H. was supported by National Institutes of Health award F32 GM130113. The Texas Advanced Computing Center (TACC) at The University of Texas at Austin provided high-performance computing resources.

Reviewers' comments:

Reviewer's Responses to Questions

**Comments to the Author**

1. Is the manuscript technically sound, and do the data support the conclusions?

Reviewer #1: Yes

Reviewer #2: Yes

2. Has the statistical analysis been performed appropriately and rigorously? 

Reviewer #1: Yes

Reviewer #2: Yes

3. Have the authors made all data underlying the findings in their manuscript fully available?

Reviewer #1: Yes

Reviewer #2: Yes

4. Is the manuscript presented in an intelligible fashion and written in standard English?

Reviewer #1: Yes

Reviewer #2: Yes

5. Review Comments to the Author

Reviewer #1: This work provides a computational approach for producing timed patterns of gene expression. In nature, timed gene expression has been observed in numerous organisms, and is particularly well characterized within bacteriophage. The authors thus use bacteriophage genes as a template, using several rounds of evolutionary simulations to produce the desired expression patterns. While expression of only a small number of genes are considered for most of this work, authors show that their approach can be scaled up to consider a larger number of genes. Notably, while many phage use specialized proteins to regulated timed gene expression (such as T7 RNA polymerase or C1 repressor), the authors show that timed expression can be achieved without the use of such proteins. Overall, I would recommend the publication of this work with minor changes. See below for additional comments.

1. A weakness in this work stems from the fact that protein translation levels were not considered. As noted in the manuscript’s introduction, ribosome binding site strength can modulate the rate of protein translation initiation by multiple orders of magnitude. Additionally, codon usage within a gene can also modulate the rate of translational elongation. Given that these translational parameters would need to be optimized within the same sequence space as RNase cleavage sites, it would likely be necessary to also consider protein translation before the computational results described in this work could be replicated within the lab. The authors should expand their discussion to include how these parameters could be accommodated by their model.

2. Particularly interesting are expression patterns shown in Figure 3, in which Gene A initially has higher expression level compared to Gene B, but later in the simulation Gene A is surpassed by expression of Gene B. This pattern seems to arise Gene A having a faster initial rate of expression, but an earlier plateau. Would it be possible to devise a configuration where Genes A, B, and C to each predominate at different time points? This would be useful for the authors to show, as it would demonstrate that the different expression patterns are more complex that simply “stronger” vs “weaker” expression.

3. Additional commentary regarding how initial expression rates versus the level at which a curve plateaus are differentially controlled would of benefit to this manuscript. For instance, what parameters lead to fast initial expression, and a low level plateau? What parameters result in a slow initial expression level, and a high level plateau?

Reviewer #2: In this manuscript, Shah et al. use their previously developed model, Pinetree, to engineer phage genomes in silico and predict the evolution of specific phenotypes. In doing so they show the ability to encode for dynamic gene expression without the use of multiple transcription factors, but rather by varying promoter, terminator, and RNA cleavage sites. They stochastically simulated the addition and removal of such sites while selecting for mutations that more closely match the desired pattern. The models they use and stochastic simulations allow for rapid evolution towards varying gene expression patterns of up to ten genes. In addition, they briefly discuss how these genomic architectures relate to those found in bacteriophages in nature. The work is novel to my knowledge, interesting, and well carried out. With a few modifications listed below and more discussion about the relevance to bacteriophage genomes and phenotypes, I recommend it for publication in PLOS One.

Major Points

1. In the results section, can you briefly relate the evolved architectures to their prevalence in bacteriophages in nature? For example, is it common to see a terminator between a promoter and its gene? You discuss this broadly in the discussion (line 393) but would be interesting to compare more specifically your evolved phenotypes.

a. Are the target patterns based on natural pathways and if so, can you compare that genomic architecture to your evolved ones?

Minor Edits

1. In Figure 1A explain the use of the symbols in the genetic diagrams and if consistent throughout the remaining figures in the paper.

a. Also explain these final genomic architectures in the figure caption(s) or main text for the evolved phenotypes (i.e. the final pattern genotype has promoters in front of each gene and terminators at X positions…)

2. Do the three small grey squares at the bottom of Fig.4 panels A&B mean anything?

3. In Figs S3 and S4, what is the transcription regulation of the 3 genes being varied in order? Is it the same for all orders of the genes for each pattern?

a. How does Fig S3 differ from the first row of graphs presented in Fig S4?

6. PLOS authors have the option to publish the peer review history of their article (what does this mean?). If published, this will include your full peer review and any attached files.

Reviewer #1: **Yes: **Corwin Miller

Reviewer #2: **Yes: **Razan N Alnahhas

---

## [Author Response · Author response to Decision Letter 0]

26 Mar 2022

This re-submission includes an extensive "response to reviewers" document upload, which is well formatted for clarity. Here, we will copy/paste for simplicity:

Response to Reviewer Comments

We would like to thank both reviewers for their overall positive evaluation of our work and for their constructive comments. We have made every effort to address them in the revision. For a detailed, point-by-point response, please see below. Reviewer text is typeset in blue, and our responses are typeset in black.

Reviewer 1

This work provides a computational approach for producing timed patterns of gene expression. In nature, timed gene expression has been observed in numerous organisms, and is particularly well characterized within bacteriophage. The authors thus use bacteriophage genes as a template, using several rounds of evolutionary simulations to produce the desired expression patterns. While expression of only a small number of genes are considered for most of this work, authors show that their approach can be scaled up to consider a larger number of genes. Notably, while many phage use specialized proteins to regulated timed gene expression (such as T7 RNA polymerase or C1 repressor), the authors show that timed expression can be achieved without the use of such proteins. Overall, I would recommend the publication of this work with minor changes. See below for additional comments.

Thank you for your overall positive evaluation of our work.

1. A weakness in this work stems from the fact that protein translation levels were not considered. As noted in the manuscript’s introduction, ribosome binding site strength can modulate the rate of protein translation initiation by multiple orders of magnitude. Additionally, codon usage within a gene can also modulate the rate of translational elongation. Given that these translational parameters would need to be optimized within the same sequence space as RNase cleavage sites, it would likely be necessary to also consider protein translation before the computational results described in this work could be replicated within the lab. The authors should expand their discussion to include how these parameters could be accommodated by their model.

You are correct, there is an additional level of control that living organisms have access to, at the level of translation, which we did not consider here. We believe we had already covered it, in our paragraph about future research directions (2nd to last paragraph in Discussion), but to strengthen this point we have now also added the following text in the Discussion, after the paragraph starting with “While our results provide insight into…”:

However, in any biological application, we would ultimately want to manipulate protein abundances, and we have here only considered transcriptional regulation. Protein abundances are determined both by transcript abundances and by additional parameters relevant at the translation stage, such as the rate of translation initiation or the speed of translation elongation, which may be modulated by codon usage or mRNA secondary structure (Novoa & Ribas do Pouplana 2012). These additional parameters would have to be considered in our simulation model to produce realistic predictions. We note that the Pinetree simulator we used here already has the capability to model translation, by tracking the movement of individual ribosomes along transcripts (Jack & Wilke 2019). It would be straightforward to evolve systems towards target protein abundances rather than transcript abundances, and to allow mutations in translation initiation rates and/or in codon usage bias, so that each gene can evolve unique translation initiation and elongation rates.

New reference:

E. M. Novoa, L. Ribas de Pouplana (2012). Speeding with control: codon usage, tRNAs, and ribosomes. Trends in Genetics 28:574-581. https://doi.org/10.1016/j.tig.2012.07.006

2. Particularly interesting are expression patterns shown in Figure 3, in which Gene A initially has higher expression level compared to Gene B, but later in the simulation Gene A is surpassed by expression of Gene B. This pattern seems to arise Gene A having a faster initial rate of expression, but an earlier plateau. Would it be possible to devise a configuration where Genes A, B, and C to each predominate at different time points? This would be useful for the authors to show, as it would demonstrate that the different expression patterns are more complex that simply “stronger” vs “weaker” expression.

We agree it would be useful to find a configuration where genes A, B, and C predominate at different time points, but we have been unable to do so. In fact, even patterns with a single crossover event (as e.g. in Fig. 2, right-most case) were not that easy to evolve. This doesn’t mean that such patterns are impossible to generate, it only implies that they are rare among the possible range of genome architectures in our model system.

In the Discussion, we have revised the paragraph starting with “Our findings here highlight” to better address this point:

Our findings here highlight some of the strengths and the limitations of regulatory approaches built solely off varying the strength of transcription, termination, and transcript degradation. The target phenotypes we used were fairly simple. We initially only considered cases in which each gene’s transcript abundance increased linearly. We then gradually increased the complexity of each gene’s expression pattern by allowing some genes to achieve steady state at certain time points. Such patterns could be evolved but we only observed a few successful simulations among 50 replicates. Our system shows that gene products can either accumulate at different rates over time via variable strength transcription or plateau at different levels by tuning degradation rates. In the most complex cases that we investigated, a combination of variable production and degradation rates was sufficient to produce expression dynamics whereby the relative ordering of genes by abundance could switch over time (Fig. 3, patterns 7-10).

3. Additional commentary regarding how initial expression rates versus the level at which a curve plateaus are differentially controlled would of benefit to this manuscript. For instance, what parameters lead to fast initial expression, and a low level plateau? What parameters result in a slow initial expression level, and a high level plateau?

We have addressed this comment by adding a new subsection in results that summarizes how different patterns are generated by certain genome architectures. This subsection is called “Summary of observed design patterns”:

From the entirety of our simulation results, we have identified the following patterns of regulatory control. In general, combinations of promoters and terminators only lead to a linear increase of transcript abundances, possibly at differing rates, as the effect of these regulatory elements does not depend on the current abundance of a transcript. For instance, for the first gene to be expressed more rapidly than the second gene, there can be a strong promoter in front of the first gene and a weak terminator in front of the second gene. Examples include the first column of Fig. 2 and also Fig. 5. By contrast, a weak promoter in front of the first gene and a second weak promoter in front of the second gene will create a pattern where the second gene is expressed more rapidly than the first gene (second column of Fig. 2). RNase cleavage sites are required for time courses where transcript levels reach a steady state, as the amount of degradation following cleavage is proportional to transcript levels and thus increases with increasing transcript abundance (examples include the three right-most columns in Fig. 2 and all genome architectures in Fig. 4B).

For time courses with a cross-over event, where one transcript levels off while another, initially less abundant transcript keeps steadily increasing (last column in Fig. 2), we have found that typically a weak promoter precedes the first gene, a strong promoter precedes the second gene, and an RNase cleavage site also precedes the second gene but follows the second promoter. The weak promoter allows for the first gene to steadily increase at a low rate. The strong promoter rescues transcription of the downstream gene. The following RNase cleavage site will degrade the transcripts produced, in proportion to the transcripts’ abundance, and thus the transcripts of the second gene will eventually reach steady state.

Finally, even though RNase cleavage sites will generally cause eventual steady state, we have also observed cases where a weak RNase cleavage site follows a strong promoter, resulting in a time course that does not meaningfully level off on the time scale of our simulation results (examples include the third, fourth, and fifth genome architecture in Fig. 4A). In those cases, the RNase cleavage site simply serves to reduce the effect of the promoter. This observation highlights that the transcript level at which a steady state is reached depends on the relative strengths of the promoter and the RNase cleavage site, and for sufficiently strong promoters and weak cleavage sites a steady state may never be observed during physiological simulation times, even though it exists and would be reached if we let the simulations run for sufficiently long times.

Reviewer 2

In this manuscript, Shah et al. use their previously developed model, Pinetree, to engineer phage genomes in silico and predict the evolution of specific phenotypes. In doing so they show the ability to encode for dynamic gene expression without the use of multiple transcription factors, but rather by varying promoter, terminator, and RNA cleavage sites. They stochastically simulated the addition and removal of such sites while selecting for mutations that more closely match the desired pattern. The models they use and stochastic simulations allow for rapid evolution towards varying gene expression patterns of up to ten genes. In addition, they briefly discuss how these genomic architectures relate to those found in bacteriophages in nature. The work is novel to my knowledge, interesting, and well carried out. With a few modifications listed below and more discussion about the relevance to bacteriophage genomes and phenotypes, I recommend it for publication in PLOS One.

Thank you for your overall positive evaluation of our work.

Major Points

1. In the results section, can you briefly relate the evolved architectures to their prevalence in bacteriophages in nature? For example, is it common to see a terminator between a promoter and its gene? You discuss this broadly in the discussion (line 393) but would be interesting to compare more specifically your evolved phenotypes.

We don’t have easy access to a systematic accounting of the different architectures in nature, as it would require careful experimental study of each individual phage genome. Therefore, we now describe genome architectures of three example phages, which should serve as sufficient motivation for our study. In fact, all three phages display the types of regulatory patterns that we evolved in our simulations.

We have added the following text as the third paragraph in the Discussion:

The arrangement of regulatory elements that arose in our simulations look similar to what is observed in bacteriophage genomes. For example, the T7 genome contains 10 RNase III cleavage sites, six of which co-occur in the same intergenic region as a promoter (Dunn and Studier, 1983). This arrangement of promoters and cleavage sites results in transcript ramps and cliffs (Jack et al., 2019), where transcript abundances fall off near cleavage sites, and then steadily increase due to the presence of promoters. Similarly, bacteriophage phiX174 has a small (5386 nt) circular genome composed of 11 genes, including a cluster of co-localized capsid genes (Sanger et al., 1977). Capsid gene expression is regulated by four terminators of varying strengths (Hayashi et al., 1981), arranged in an alternating gene-terminator pattern similar to that of the evolved ten-gene architecture (Fig. 5). In phiX174, terminator readthrough produces a stepwise pattern of decreasing transcript abundances (Logel and Jaschke, 2020), which may help ensure that structural proteins are produced with the stoichiometries necessary for proper capsid assembly. PhiX174 also contains an overlapping promoter-terminator element, situated in the intergenic region between its structural and replication genes. In our evolved genome architectures, there is at least one example of such an element (Fig. 1A, right two panels); in both our simulation and in phiX174, this regulatory feature appears to decouple expression of the adjacent genes. Finally, bacteriophage T4, which has a large genome encoding over 300 genes, makes use of at least 119 promoters, as well as a number of terminators and mRNA cleavage sites, many of which co-occur as groups of two or more regulatory elements (Miller et al., 2003). Combinations of promoters, terminators, and cleavage sites give rise to the complex, time-dependent gene expression patterns observed in T4, which are further controlled by phage-encoded regulatory proteins.

References in order of appearance:

Dunn and Studier, 1983: https://doi.org/10.1016/S0022-2836(83)80282-4

Jack et al., 2019: https://doi.org/10.1093/ve/vez055

Sanger et al., 1977: https://doi.org/10.1038/265687a0

Hayashi et al., 1981: https://doi.org/10.1128/jvi.38.1.198-207.1981

Logel and Jaschke, 2020: https://doi.org/10.1016/j.virol.2020.05.008

Miller et al., 2003: https://doi.org/10.1128/MMBR.67.1.86-156.2003

a. Are the target patterns based on natural pathways and if so, can you compare that genomic architecture to your evolved ones?

No, the target patterns were not based on natural pathways. We simply devised a range of basic patterns that should be useful in principle. See also our response to Point 2 of Reviewer 1, where we talk about the topic of target pattern selection in more detail.

Minor Edits

1. In Figure 1A explain the use of the symbols in the genetic diagrams and if consistent throughout the remaining figures in the paper.

Symbols are used consistently throughout the manuscript. To clarify what they mean, we have added the following text to the caption of Fig 1:

This architecture has a promoter in front of genes X and Y (indicated by inverted L with an arrow pointing right), a transcriptional terminator after genes Y and Z (indicated by a T), and an RNase cleavage site in front of gene Y (indicated by a cross). Line colors correspond to the genes labeled in the genome architecture. We use these symbols and colors consistently throughout this work to display genome architectures. 

a. Also explain these final genomic architectures in the figure caption(s) or main text for the evolved phenotypes (i.e. the final pattern genotype has promoters in front of each gene and terminators at X positions…)

We have done so where feasible. It would be repetitive to enumerate every single architecture, but we agree that more extensive descriptions of genome architectures are useful. For an example of the revisions made, see our response to the previous point. Also see the new subsection we have added to the Results, in response to Point 3 of Reviewer 1.

2. Do the three small grey squares at the bottom of Fig.4 panels A&B mean anything?

Yes, they are an ellipsis, meant to indicate that there were additional architectures that aren’t shown, as in: A, B, C, … 

In hindsight, we agree that the figure was a bit confusing. We have revised the figure so that now the dots are arranged vertically. That should make it more immediately obvious that the dots are an ellipsis. We have also clarified in the figure caption that there are additional genome architectures that are not shown, by adding the following text:

Only the architectures that evolved most frequently are shown. Multiples on the right indicate how often each architecture arose among 50 replicates. 

3. In Figs S3 and S4, what is the transcription regulation of the 3 genes being varied in order? Is it the same for all orders of the genes for each pattern?

a. How does Fig S3 differ from the first row of graphs presented in Fig S4?

First, we’d like to emphasize that Figs S3 and S4 are entirely different and serve different purposes. Fig S3 shows different possible target patterns, so the concept of regulatory architecture doesn’t apply there. Its purpose is to show that gene identity matters, and that we need to take this into account in our analysis.

The first row of Fig S4 shows that the pattern highlighted in Fig S3 can be evolved for all gene orders. Other rows in Fig. S4 show that the same is (broadly) true for the other target patterns considered. In general, they can be matched regardless of gene order. We have revised the caption of Fig S4 to explain this point more clearly.

---

## [Editor Report · Decision Letter 1]

11 May 2022

Generating dynamic gene expression patterns without the need for regulatory circuits

PONE-D-21-23762R1

Dear Dr. Hockenberry,

We’re pleased to inform you that your manuscript has been judged scientifically suitable for publication and will be formally accepted for publication once it meets all outstanding technical requirements.

Kind regards,

William Ott, Ph.D.

Academic Editor

PLOS ONE

Additional Editor Comments (optional):

The authors have elegantly addressed all of the points raised by Referees 1 and 2.
---

## [Editor Report · Acceptance letter]

18 May 2022

PONE-D-21-23762R1 

Generating dynamic gene expression patterns without the need for regulatory circuits 

Dear Dr. Hockenberry:

I'm pleased to inform you that your manuscript has been deemed suitable for publication in PLOS ONE. Congratulations! Your manuscript is now with our production department. 

Kind regards, 

on behalf of

Dr. William Ott 

Academic Editor

PLOS ONE